# Effect of Two Different Sperm Selection Methods on Boar Sperm Parameters and In Vitro Fertilisation Outcomes

**DOI:** 10.3390/ani14172544

**Published:** 2024-09-01

**Authors:** Maria Serrano-Albal, Marie Claire Aquilina, Lucas G. Kiazim, Louisa J. Zak, Darren K. Griffin, Peter J. Ellis

**Affiliations:** 1School of Biosciences, University of Kent, Canterbury CT2 7NZ, UK; 2Topigs Norsvin Research Center, Meerendonkweg 25, 5216 TZ ‘s-Hertogenbosch, The Netherlands

**Keywords:** sperm morphology, sperm motility, microfluidics, density gradient centrifugation, embryo production

## Abstract

**Simple Summary:**

This study compares two methods of preparing boar sperm for in vitro fertilization: the traditional density gradient selection (DGS) using centrifugation and the newer microfluidic chip-based sperm (MCS) sorting. MCS resulted in lower sperm concentration and fewer morphologically abnormal sperm compared to DGS. Additionally, although DGS showed higher progressive motility, both methods had similar fertilization outcomes in terms of cleavage rates, blastulation rates, and embryo quality. The results show that MCS performs at least as well as DGS and may offer advantages due to its gentler approach and potential for more consistent results in IVF procedures.

**Abstract:**

Porcine in vitro embryo production (IVP) protocols have conventionally used density gradient selection (DGS) by centrifugation to prepare sperm samples and achieve successful fertilisation. However, the possible toxicity of the solutions used and the potential damage caused by the centrifugation step may have a negative effect on the quality of the sample. Microfluidic chip-based sperm (MCS) sorting has been proposed as an alternative technique for the selection of high-quality sperm with the purpose of improving reproductive outcomes in IVF. This device does not require centrifugation or any toxic solution to prepare the sample for fertilisation. The sample is not subjected to unnecessary stress, and the process is less operator-dependent. In this study, we compared the sperm parameters of unselected extender-diluted boar semen samples with selected samples using DGS and MCS methods. The results show an expected reduction in sperm concentration after both methods. All the groups were significantly different from one another, with MCS being the group with the lowest concentration. Though the three groups had a similar overall motility, significant differences were found in progressive motility when comparing the unselected group (control, 19.5 ± 1.4%) with DGS and MCS. Progressive motility in DGS was also significantly higher than in MCS (65.2 ± 4.9% and 45.7% ± 5.3, respectively). However, MCS selection resulted in enriched sperm samples with a significantly lower proportion of morphologically abnormal sperm compared to DGS. After fertilisation, no statistical differences were found between the two methods for embryological parameters such as cleavage rates, blastulation rates, and embryo quality. The number of cells in blastocysts derived from MCS was significantly greater than those derived from DGS sperm. Thus, we demonstrate that MCS is at least as good as the standard DGS for most measures. As a more gentle and reproducible approach for sperm selection, however, it could improve consistency and improve IVP outcomes as mediated by a greater proportion of morphologically normal sperm and manifested by a higher cell count in blastocysts.

## 1. Introduction

The expansion in global pork consumption has driven the pig breeding industry as a whole to enhance production in a more efficient and environmentally sustainable manner [1]. This is being achieved by the combination of genetics and assisted reproduction technology (ART). However, the use of ART in swine is limited to artificial insemination (AI) alone. This is mostly due to the technical difficulties in producing pig in vitro embryos of comparable quality to in vivo-produced embryos and the inability to achieve suitable pregnancy rates after transfer [2,3].

Sperm quality is an essential element in the success of in vitro fertilisation (IVF) and embryo development in humans [4], cattle [5], and pigs [6,7,8]. It is commonly indicated by motility, viability, morphology, and concentration in the ejaculate. Each boar ejaculate compiles a diverse population of sperm with varied degrees of morphological and functional differentiation and normality [9,10]. To isolate only the sperm populations with the highest potential for fertilisation, sperm selection methods have been proposed to optimise IVF [11,12].

In vivo, boar sperm go through a rigorous selection process during their transit through the female reproductive tract. They undergo biological changes, known as capacitation [13], that cause morphological and structural changes in the sperm [14]. They also are filtered out into high-quality sperm from the remainder of the ejaculate by active migration across the cervical mucus [15,16]. This favours very efficient fertilisation [17], where around 37 billion are released into the female genital tract but only around 5000 are able to reach the fertilisation site [18]. Sperm selection methods for IVF have been designed to mimic this process. Establishing the best method for sperm selection prior to IVF presents several challenges. Ideally, this method should be non-invasive and cost-effective. It should isolate as many motile sperm cells as possible without causing damage and be able to identify high-quality sperm to obtain better outcomes in terms of pregnancy and live birth rates [19,20].

There are different approaches for sperm separation that conventionally involve centrifugation such as the simple wash method, the swim-up method, and density gradient selection (DGS) [21,22,23]. Density gradients, using solutions with colloidal silica particles, are widely used to select superior-quality sperm by centrifugation in many livestock species such as bull [24] and boar [25,26,27]. In porcine in vitro production (IVP), DGS is the gold standard method, since it is able to enrich the quality of the boar semen samples with respect to morphology and motility [28], showing an improvement in penetration [27], cleavage [29], and blastulation rates [29,30] after IVF and embryo culture. However, the centrifugation process and the contact of the sperm with the solutions used in DGS have potential adverse effects on sperm. They may cause mechanical damage [21,24,31] and increase oxidative stress levels, which may result in reduced spermatozoa motility and reduce the spermatozoa’s ability to bind to oocyte membranes. This, in turn, may weaken the fertilisation process and compromise embryo development [32,33,34].

In recent years, microfluidic-based sperm selection (MCS) devices have been proposed as the most promising of the emerging sperm selection techniques [23,35,36,37]. These devices are a simple and low-cost alternative to conventional sperm selection that perform a less invasive separation, relying on the behaviour and movement of sperm in the absence of any external stimuli, simulating the cervical and uterine pathways that sperm must cross to naturally fertilise the mature oocyte [37,38]. In humans, MCS seems to result in preparations with a reduced proportion of spermatozoa with DNA fragmentation [39,40,41]; however, in livestock species, there are few studies using MCS devices. In 2018, Nagata et al. [42] published the improvement in selecting bull sperm using a diffuser-type microfluidic sperm sorter; however, this approach has not been studied with boar sperm, nor using a device like the ZyMōt™ Multi, (ZyMot Fertility, Gaithersburg, USA) which is mostly used in human IVF clinics, showing an increase in the euploidy rates in IVF embryos [41,43,44]. ZyMōt™ Multi has also been tested recently in equine IVP, yielding substantially higher-quality sperm than using DGS [45].

The aim of this study was thus to test the hypothesis that sorting sperm using a commercial microfluidic device improves the selection of boar sperm (as measured by standard andrology) and improves pig IVF outcomes compared to standard methods.

## 2. Materials and Methods

All chemicals and reagents used were bought from Sigma-Aldrich (Gillingham, UK) unless otherwise stated. For the analysis of the sperm parameters, boar semen samples were divided into three groups: unselected semen in extender (control), sperm selection by the microfluidic sperm sorting (MCS), and sperm selection by sperm density gradient selection (DGS).

### 2.1. Boar Sperm Preparation and Capacitation

Extended boar semen (for commercial AI) was supplied by JSR Genetics Ltd. (Southburn, UK). The AI samples were shipped at room temperature (RT) in the post, then preserved at 17 °C in a cool box for up to two days before use.

#### 2.1.1. Sperm Density Gradient Selection (DGS)

Sperm preparation was accomplished using a 35%/70% (*v*/*v*) discontinuous density gradient system using BoviPure solutions (Nidacon, Göthenborg, Sweden). To avoid mixing the solutions, 1 mL of solution of 35% BoviPure was carefully layered on top of 1 mL of the 70% BoviPure solution with the higher density in a 15 mL conic centrifuge tube. Next, 1 mL aliquot of extended boar sperm was added on top. Once all the layers were in the tube, the samples were centrifuged for 15 min at 400 g. Afterwards, all the supernatant was removed with care to not disturb the sperm pellet and 1 mL of BoviWash (Nidacon, Göthenborg, Sweden) wash was added on top, prior to another 5 min centrifugation at 400 g. Finally, once this supernatant was removed, the pellet was resuspended in 2 mL of Porcine Gamete Medium (PGM; 108 mM NaCl, 10 mM KCl, 0.35 mM KH_2_PO_4_, 0.4 mM MgSO_4_, 25 mM NaHCO_3_, 5.0 mM glucose, 0.2 mM sodium pyruvate, 2.0 mM calcium lactate, 2.5 Mm theophylline, 1 µM adenosine, 0.25 µM L-cysteine, 10 µg/mL gentamycin, 4 mg/mL BSA) [46].

#### 2.1.2. Microfluidics Chip-Based Sperm Separation (MCS)

A 3 mL aliquot of extended boar semen was inserted through the inlet port of the lower fraction of the MCS device (Figure 1), ZyMōt™ Multi (3 mL) (ZyMōt Fertility, Inc., Gaithersburg, MD, USA). Then, 2 mL of PGM was rapidly added on top of the membrane, and the device was incubated at room temperature (RT) for 30 min. Following incubation, 1 mL of the upper fraction was aspirated gently via the outlet port. This fraction contains the higher motility sperm that were able to swim up through the membrane.

### 2.2. Boar Sperm Evaluation

The following tests for semen assessment were performed in the sperm samples before (control) and after sperm selection: total and progressive motility and other kinetic parameters, sperm morphology, sperm concentration, sperm viability, acrosome reactivity, and sperm DNA damage levels.

#### 2.2.1. Sperm Morphology

Prior to morphological evaluation, sperm samples were fixed in 4% paraformaldehyde (PFA) for 30 min at 4 °C. Then, they were visualised without any staining, using the brightfield of an Olympus BX61 epifluorescence microscope (Olympus, Tokyo, Japan), equipped with a cooled CCD camera at ×200 total magnification, and using the software SmartCapture (version 3, Digital Scientific UK, Cambridge, UK). Based on their morphology, sperm were classified as follows: normal sperm, sperm with tail abnormalities, sperm with head abnormalities, or sperm with cytoplasmic drops.

#### 2.2.2. Total Motility

Motility was assessed using the iSperm Swine Semen Analysis System (GenePro Inc., Fitchburg, WI, USA), as per the manufacturer’s instructions. The motion parameters studied were total motility (%), progressive motility (%), curvilinear velocity (VCL, μm/s), straight-line velocity (VSL, μm/s), average path velocity (VAP, μm/s), linearity of the curvilinear trajectory (LIN, ratio of VSL/VCL, expressed as percentage), and straightness (STR, ratio of VSL/VAP, expressed as percentage).

#### 2.2.3. Sperm Concentration

The sperm count of each sample was performed on a Makler chamber via phase-contrast microscopy at ×200 total magnification, using an Olympus BX61 epifluorescence microscope, equipped with a cooled CCD camera. Then, 10 µL of the sperm dilution was placed in the Makler chamber, and with this formula [(n/10) × 10^6^ sperm cells/mL) × 2], where n is the number of counted sperm, the concentration was calculated and presented as million/mL. For IVF, the desired sperm concentration was 10^5^ sperm cells/mL.

#### 2.2.4. Sperm Viability and Acrosomal Integrity

Sperm viability was assessed using propidium iodide (PI) and acrosome status was determined using fluorescein isothiocyanate-labelled peanut agglutinin (FITC-PNA; Invitrogen™, Inchinnan, UK), as explained by Robles and Martinez-Pastor [47]. Briefly, sperm samples were diluted in a staining solution containing 1.5 μM PI and 1 μg/mL of FITC-PNA (diluted in PBS) to achieve a concentration of 1–2 × 10^6^ sperm cells/mL. The samples were then analysed using a BD Accuri C6 Plus flow cytometer (BD Biosciences, Wokingham, Berkshire, UK) equipped with a 14.7 mW 640 nm Diode Red Laser and 20 mW 488 nm Solid State Blue Laser. The BD Accuri C6 Software v.1.34.1 was used for data processing and a total of 10,000 events were collected per sample. Fluorescence of PNA was detected using a 533/30 nm bandpass filter (FL1) and fluorescence of PI was detected using a 670 nm bandpass filter (FL3).

#### 2.2.5. Sperm DNA Damage Testing

The sperm DNA damage levels were assessed using acridine orange dye based on a previously published protocol [48]. Briefly, sperm cells were diluted in a TNE buffer solution and then treated with an acid detergent solution for 30 s. Acridine orange stain was added, which binds differently to double-stranded DNA (green fluorescence) versus single-stranded DNA (red fluorescence) due to its metachromatic properties. The acridine orange stain also allowed for the quantification of spermatozoa with incomplete chromatin condensation, characterised by a relatively high green, fluorescent intensity and referred to as high DNA stainability (HDS). The samples were analysed using a BD Accuri C6 Plus flow cytometer (BD Biosciences, Wokingham, Berkshire, UK), and the BD Accuri C6 Software v.1.34.1 was then employed to calculate the percentage of DNA fragmentation index (%DFI) and the percentage of high DNA stainability (%HDS). Triplicate measurements were taken per sample with a total of 5000 sperm cells collected per run. Data corresponding to green (FL1, 533/30 nm bandpass filter) and red (FL3, 670 nm bandpass filter) fluorescence were collected. The %DFI was derived from the ratio of red to total (red plus green) fluorescence intensity. The %HDS was calculated by measuring the proportion of sperm showing green fluorescence intensity exceeding the upper limit of the main sperm population cluster. 

### 2.3. Porcine Oocyte Collection and in Vitro Maturation (IVM)

Prepubertal gilt ovaries were collected from C & K Meats Limited (Suffolk, UK) and transported to the laboratory within 3 h in a sealed bag submerged in water at 30–35 °C. At slaughter, the animals weighed approximately 160 kg, and their prepubertal status was confirmed by the absence of developed ovarian corpora lutea. Before aspiration, the ovaries were washed two to three times in 1× PBS and kept at 28 °C in a water bath. The retrieval of cumulus–oocyte complexes (COCs) was performed by manual aspiration from non-atretic follicles (3–6 mm) using a non-pyrogenic/non-toxic syringe (Henke-Sass Wolf GmbH, Tuttlingen, Germany) fitted with an 18-gauge needle. The follicular fluids were washed three times in a modified HEPES-buffered Porcine X Medium (PXM [46]; 108 mM NaCl, 10 mM KCl, 0.35 mM KH_2_PO_4_, 0.40 mM MgSO_4_, 5.0 mM NaHCO_3_, 25 mM HEPES, 0.2 mM sodium pyruvate, 2.0 mM calcium lactate, 4mg/mL bovine serum albumin (BSA)), warmed at 38 °C.

The chosen COCs were washed three times in a modified Porcine Oocyte Medium (POM [49]; 108 mM NaCl, 10 mM KCl, 0.35 mM KH_2_PO_4_, 0.4 mM MgSO_4_, 25 mM NaHCO_3_, 5.0 mM glucose, 0.2 mM sodium pyruvate, 2.0 mM calcium lactate, 2.0 mM glutamine, 5.0 mM hypotaurine, 0.1 mM cysteine, 20 µL/mL BME amino acids 50×, 10 µL/mL MEM non-essential amino acids 100×, 10 ng/mL EGF, 50 µM β-mercaptoethanol, 10 µg/mL gentamycin, 4 mg/mL BSA, 40 ng/mL FGF2, 20 ng/mL LIF, and 20 ng/mL IGF) previously equilibrated overnight at 38.5 °C and 5.5% CO_2_ in humidified air. COCs were randomly distributed into groups of 50 oocytes for maturation. During the first 20 h of culture in POM, COCs were supplemented with FSH (0.5 IU/mL), LH (0.5 IU/mL), and dbc-AMP (0.1 mM). During the subsequent 24 h, oocytes were cultured in POM with the different studied supplements but in the absence of hormones and dbc-AMP at 38.5 °C in a saturated humidity atmosphere of 5.5% CO_2_ in the air.

### 2.4. Porcine in Vitro Fertilisation and Embryo Culture

After in vitro maturation (IVM), matured oocytes were washed twice in a modified Porcine Gamete Medium (PGM [49]; 108 mM NaCl, 10 mM KCl, 0.35 mM KH_2_PO_4_, 0.4 mM MgSO_4_, 25 mM NaHCO_3_, 5.0 mM glucose, 0.2 mM sodium pyruvate, 2.0 mM calcium lactate, 2.5 Mm theophylline, 1 µM adenosine, 0.25 µM L-cysteine, 10 µg/mL gentamycin, 4 mg/mL BSA 40 ng/mL FGF2, 20 ng/mL LIF, and 20 ng/mL IGF). Then, oocytes were co-incubated with the prepared sperm by MCS or DGS for two hours. To reduce the risk of polyspermy, oocytes were then moved to a clean well of PGM to reduce the concentration of sperm, for another two hours.

Hereafter, presumptive zygotes were denuded by pipetting up and down for 30 s in the well and washed twice in a modified Porcine Zygote Medium 5 (PZM5 [49]; 108 mM NaCl, 1 mM KCl, 0.35 mM KH_2_PO_4_, 0.4 mM MgSO_4_, 25 mM NaHCO_3_, 0.2 mM sodium pyruvate, 2.0 mM calcium lactate, 2.0 mM glutamine, 5.0 mM hypotaurine, 20 µL/mL BME amino acids 50×, 10 µL/mL MEM non-essential amino acids 100×, 10 µg/mL gentamycin, 4 mg/mL BSA, 40 ng/mL FGF2, 20 ng/mL LIF, and 20 ng/mL IGF). The culture was performed in 500 µL wells of PZM5 overlaid with mineral oil and plates were incubated for five days at 38.5 °C, in 5.5% CO_2_ and 6% O_2_ in humidified air.

### 2.5. Evaluation of Pig Blastocysts 

Blastocyst morphological appearance was assessed and scored using three grades: (1) excellent—fully expanded blastocyst, spherical, regular border, symmetrical with uniform size cells, obvious inner cell mass (ICM), and densely populated trophectoderm (TE); (2) good—expanded blastocyst with few small blastomeres, fewer cells forming the ICM/TE; (3) poor—expanded or less developed blastocyst with numerous extruded blastomeres, loosely populated TE, and possibly ICM. After grading, blastocysts from each treatment group were fixed in 4% PFA for 30 min at 4 °C and stained with Hoechst H3570 (Invitrogen™) for cell counts. Blastocysts were visualised under an Olympus BX61 epifluorescence microscope as described in the previous section.

### 2.6. Statistical Analysis

SPSS was used to analyse the data (Version 25, SPSS Inc., Chicago, USA). The samples were statistically analysed using one-way ANOVA in those parameters with a normal distribution as determined by the Kolmogorov–Smirnov Test. The Kruskal–Wallis Test was used to test the data parameters that did not have a normal distribution. Results were considered statistically significant when *p* < 0.05. Violin graphs were plotted by https://www.bioinformatics.com.cn/en, accessed on 1 November 2023, a free online platform for data analysis and visualisation. The evaluation of IVF outcomes (cleavage, blastulation rates, and embryo quality) was analysed using the chi-square test (*p* < 0.05).

## 3. Results

The use of any of the sperm selection methods reduces significantly the concentration of the sperm samples (Figure 2A); in the control group, we obtained an average concentration of 33.4 ± 2.2 million sperm cells/mL, while the concentration after using DGS was 10.1 ± 1.2 million sperm cells/mL, and 3.9 ± 0.6 million sperm cells/mL after using MCS. Both test groups were lower than the control, as expected, but the MCS group was significantly lower than DGS.

The percentage of total motility in the samples before (control: 58.8 ± 4.4%) and after the use of MCS (73.0 ± 3.8%) as sperm selection did not differ significantly (Figure 2B). The total motility after using DGS (75.9 ± 4.0%) did significantly differ from the control group but was not statistically different from MCS. The proportion of sperm with progressive motility was significantly lower in the control (19.5 ± 1.4%) than in both selection methods (Figure 2C). The differences found in the proportion of sperm with progressive motility in the groups DGS and MCS were also significant, as DGS (65.2 ± 4.9%) selected samples showed higher progressive motility than the MCS (45.7% ± 5.3) ones.

The analysis of the motion parameters obtained by the iSperm Swine Semen Analysis System showed that the curvilinear velocity of the samples (VCL, Figure 3A) significantly increased when selected with DGS (92.5 ± 3.4 μm/s) compared to the control (60.2 ± 2.7 μm/s) and MCS (71.1 ± 3.6 μm/s), MCS did not differ significantly with control. The values showed for the average path velocity (VAP, Figure 3A) and straight-line velocity (VSL, Figure 3A) of each group were significantly different from one another. Both velocity variables of MCS (VAP = 66.3 ± 3.3 μm/s and VSL = 55.1 ± 2.6 μm/s) were significantly increased compared to control (48.3 ± 2.5 μm/s and 39.6 ± 2.4 μm/s for VAP and VSL, respectively), but not as big as the increase on DGS (83.1 ± 2.1 μm/s and 75.7 ± 2.9 μm/s for VAP and VSL, respectively). The linearity (LIN, Figure 3B) of the samples from MCS (77.6 ± 2.1%) and DGS (79.9 ± 2.3%) were statistically similar, LIN in both groups was significantly higher compared to control (62.9 ± 2.1%). The straightness of trajectory (STR, Figure 3B) is significantly different in DGS (89.2 ± 1.7%) compared to control (78.6 ± 1.9%), but it did not differ with MCS (83.4 ± 2.3%). MCS did not significantly differ from the control.

The evaluation of the sperm morphology performed before (control = 29.6 ± 1.5%) and after sperm selection showed significantly lower proportions of anomalies with both selection methods (*p* < 0.01, Figure 2D). However, the percentage of abnormalities found in the sperm selected by MCS (15.2 ± 1.0%) was significantly lower than in the DGS group (24.7 ± 1.3%). Looking into the different types of abnormalities (Table 1), our results showed no significant differences in the percentage of head abnormality between all the studied groups. When looking at tail abnormalities, the percentage found in the control group (10.5 ± 1.2%) was significantly higher than in the MCS and DGS groups. MCS showed the lowest percentage of tail anomalies (4.9 ± 0.6%), differing statistically from the percentage found in DGS (8.6 ± 1.0%). Our results showed significant differences in the percentage of presence of cytoplasmic drop (CD) between all the groups. The highest percentage was found in the control group (11.5 ± 1.6%), followed by DGS (8.8 ± 1.2%). MCS showed the lowest percentage (3.6 ± 0.6%).

The testing of sperm viability (Figure 4A) and acrosome reactivity (Figure 4B) showed a significant decrease in the proportion of viable sperm after performing DGS (82.8 ± 4.6%) when compared to the control (97.1 ± 0.4%; *p* < 0.01) and the MCS groups (95.3 ± 1.0%; *p* < 0.01). There was no significant difference in the proportion of alive sperm between the MCS and control group. The proportion of acrosome-reacted sperm following DGC (11.8 ± 6.9%) was higher compared to the control (5.0 ± 1.9%) and MCS (5.0 ± 1.4%) groups; however, it was not a significant increase (*p* = 0.18).

The %DFI (Table 2) showed a significant reduction after employing both sperm separation techniques. The DFI decreased from 2.4 ± 0.3% in the original extended sample to 1.1 ± 0.2% after density gradient centrifugation and 0.7 ± 0.1% after MCS. However, there was no statistically significant difference in the DFI values between the MCS and DGS sperm preparation methods. MCS resulted in a significantly lower proportion of high %HDS spermatozoa when compared to the control group (*p* < 0.001). Additionally, the %HDS sperm was lower after DGS (1.7 ± 0.3% for DGS vs. 2.3 ± 0.4% for the control group), although the difference was not statistically significant.

Following sperm extraction from the upper chamber of the microfluidic device, sperm from the bottom chamber was also extracted to analyse the efficacy of the device in separating sperm. There was a significant difference in %DFI between the two chambers (0.7% ± 0.1% upper chamber vs 2.9% ± 1.6% bottom chamber; *p* < 0.05), indicating that sperm having lower levels of DNA damage were more likely to pass through the microfluidic channels.

When evaluating in vitro fertilisation outcomes, there are no discernible changes (*p* > 0.05) between the two sperm selection strategies, based on the proportion of embryos that cleaved (Figure 5A) and the proportion of embryos that reached blastocyst stage after 5 days of in vitro culture (Figure 5B). The cleavage rate obtained using the sperm selected by DGS was 46.7%, while the obtained using MCS-selected sperm was 42.5%. The observed blastulation rates were 11.9% and 10.4% for the DGS and MCS groups, respectively.

Morphological evaluation of the embryos that reached the blastocyst stage showed most of the blastocysts from the DGS group were classified as good quality (53.5%) or poor quality (38.5%) embryos, and only 7.8% of excellent quality. In the MCS group, the embryos obtained were of an overall higher quality, with only 27.3% being classified as poor, 36.4% for good, and 36.3% for excellent quality embryos. However, the statistical analysis showed no significant differences in the quality of the blastocyst from both groups (Figure 5C).

The statistical analysis, when comparing the mean number of cells per blastocyst produced by each sperm selection technique, revealed a significantly higher number of cells in those embryos that were fertilised with sperm selected using MCS (Figure 5D). The average number of cells per blastocyst in the MCS group was 49.6 ± 3.0 cells, while the average number of cells in the DGS group was 38.7 ± 1.8 cells (ANOVA; F_1_ = 12.389; *p* < 0.05).

## 4. Discussion

One of the changes that sperm undergoes during capacitation is hyperactivation [50,51]. This process is described as the increased beating amplitude of the sperm flagellum, resulting in a more vigorous movement, and is essential for penetrating COCs [52]. Sperm progressive motility was higher in DGS, but the linearity and the straightness of trajectory in selected samples were similar in both studied methods. To achieve better MCS motility, particularly as there is no centrifugation step to support capacitation, the addition of capacitating agents to the IVF media may be helpful, such as increasing the intracellular concentration of calcium using a calcium ionophore [50,53] or caffeine [54,55].

In IVF, the important reduction in the concentration of the samples selected by MCS might be a problem for those boars with a lower concentration or poorer quality [8]. When using DGS, the selection is less harsh, since centrifugation facilitates the accumulation of sperm, so the concentrations of the samples after using DGS tend to be higher than when using MCS. Nevertheless, the use of centrifugation can increase the formation of sperm aggregates during sperm capacitation, which might hinder the fertilisation process. The gentler handling of the sperm using MCS devices like ZyMōt™ reduces the formation of aggregates, which could positively affect IVF results. In terms of other ARTs, like ICSI, MCS would be ideal to select good sperm without capacitating the sperm considerably, facilitating the selection of the sperm by the embryologist [56]. Anderson et al. [43] showed that using MCS as a sorting sperm method for ICSI improves the number of euploid human embryos.

In relation to sperm morphology, density gradients are known to reduce the levels of abnormalities in the semen sample, eliminating dead sperm cells, sperm with head or tail anomalies, or cells with cytoplasmic droplets and debris that are known to generate a high proportion of ROS [28]. However, here we demonstrate that MCS performs a more stringent selection reducing even more the abnormality levels in the sample. Centrifugation has been reported to be detrimental to the structural integrity of sperm, reducing their lifespan and their ability to fertilise [32]. In bovine sperm, Avery and Greve [57] suggested that adverse effects on cleavage rates after in vitro insemination using density gradients could be caused by the effect of unbound polyvinylpyrrolidone in the solutions used. Interestingly, DGS was the process associated with a lower number of viable spermatozoa relative to the MCS and the control. Additionally, DGS had a higher number of acrosome-reacted spermatozoa. Such an observation could be due to the centrifugation process potentially inducing membrane instability and shortening the life span of the spermatozoa [58,59]. This can result in premature acrosome reaction and preclude zona binding.

The findings of this study indicate that both sperm preparation techniques (MCS and DGS) were effective in selecting spermatozoa with lower levels of DNA fragmentation. However, MCS was associated with the lowest DFI values, although the differences with DGS were not statistically significant. While some previous studies have suggested that the centrifugation process involved in DGS can increase levels of reactive oxygen species in the media, potentially leading to DNA damage [60], other contradictory reports have shown that DGS does not elevate reactive oxygen species or DFI levels [61,62]. Our study demonstrates a lack of significant difference in DFI levels between the DGS and MCS techniques, albeit with a trend towards lower DFI values following MCS.

In terms of fertility and embryo development, both DGS and MCS will work perfectly as selective sperm methods in pig IVF, since this study showed similar cleavage and blastulation rates. One of the problems to face in pig IVF is the high rates of polyspermy that decreases embryo development and the number of cells [63]. Even in the studied groups, there were no differences in the quality of the blastocyst, and we have reported a significantly higher number of blastomeres in embryos from the MCS group.

To conclude, even though DGS is selecting sperm with higher motility, the benefits of using less invasive selecting approaches like MCS can produce the same number of embryos with a higher number of cells, which can be translated as them having increased developmental potential.

## Figures and Tables

**Figure 1 animals-14-02544-f001:**
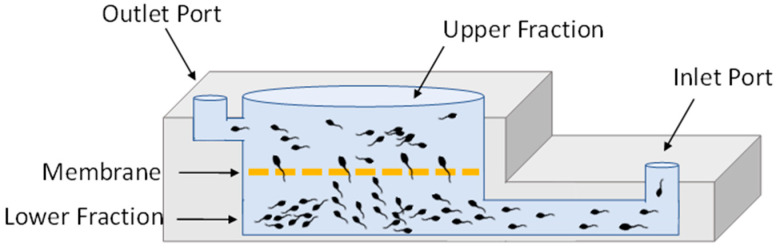
Diagram of the microfluidic sperm sorting device ZyMōt™ Multi. Device that separates the sperm based on motility. Sperm sample is inserted through the inlet port. Motile spermatozoa actively swim through the micropores within the membrane, which are then collected from the upper fraction using the outlet port.

**Figure 2 animals-14-02544-f002:**
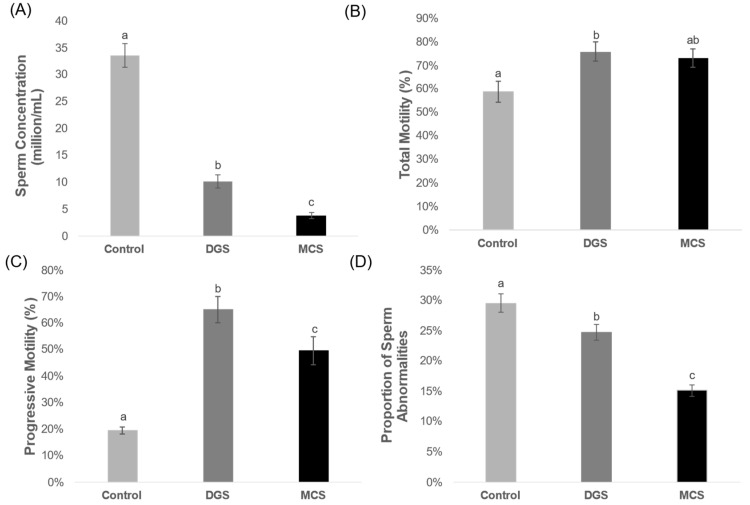
Effects of sperm selection methods (MCS and DGS) on concentration, motility, and morphology. (**A**) Effect of sperm selection methods on the final concentration of the sample. Kruskal–Wallis Test (gl = 2, *p* < 0.001, H = 40.451). (**B**) Effect of sperm selection methods on the average percentage of total motility. ANOVA Test (gl = 2, *p* = 0.010; F = 5.075). (**C**) Effect of sperm selection methods on average percentage of progressive motility. ANOVA Test (gl = 2, *p* < 0.001; F = 30.499). (**D**) Effect of sperm selection methods on proportion of sperm abnormalities from the evaluation of sperm morphology. ANOVA Test (gl = 2, *p* < 0.01, F = 32.832). For all panels, data are shown as mean% ± SEM (N = 17 replicates for each group). a, b, c Different superscript letters indicate significant differences amongst the groups; groups that share the same letter are statistically indistinguishable from each other. Abbreviations: MCS = microfluidic chip-based sperm; DGS = density gradient system.

**Figure 3 animals-14-02544-f003:**
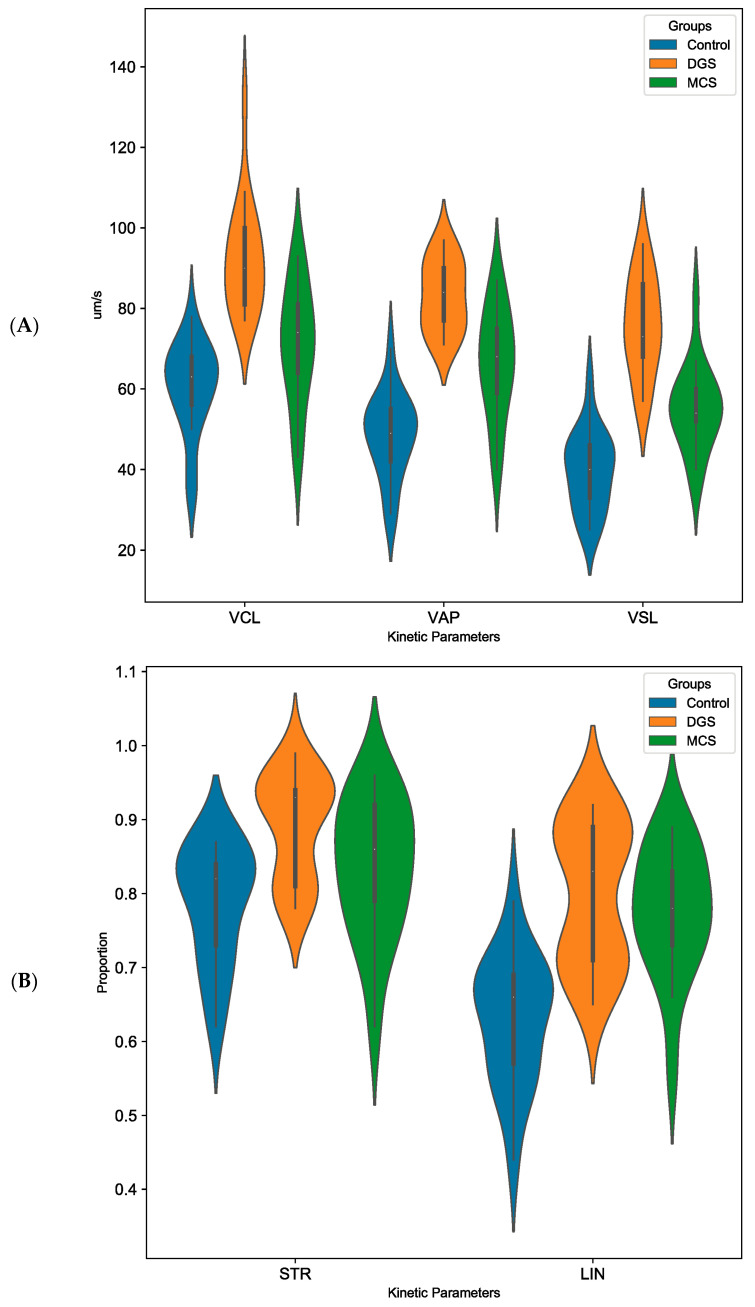
Effect of sperm selection method on sperm kinetic parameters (VCL, VSL, VAP, LIN, and STR). (**A**) From left to right: curvilinear velocity (VCL), and average path velocity (VAP), straight-line velocity (VSL). (**B**) From left to right: linearity (LIN) and straightness of trajectory (STR). N = 17 replicates for each group. ANOVA test (gl = 2, *p* value of all these parameters < 0.02). F_VCL_ = 25.557; F_VAP_ = 41.704; F_VSL_ = 46.637; F_LIN_ = 18.235; F_STR_ = 7.60 (F were calculated as variation between sample means/variation within the samples). Abbreviations: MCS = microfluidic chip-based sperm; DGS = density gradient system.

**Figure 4 animals-14-02544-f004:**
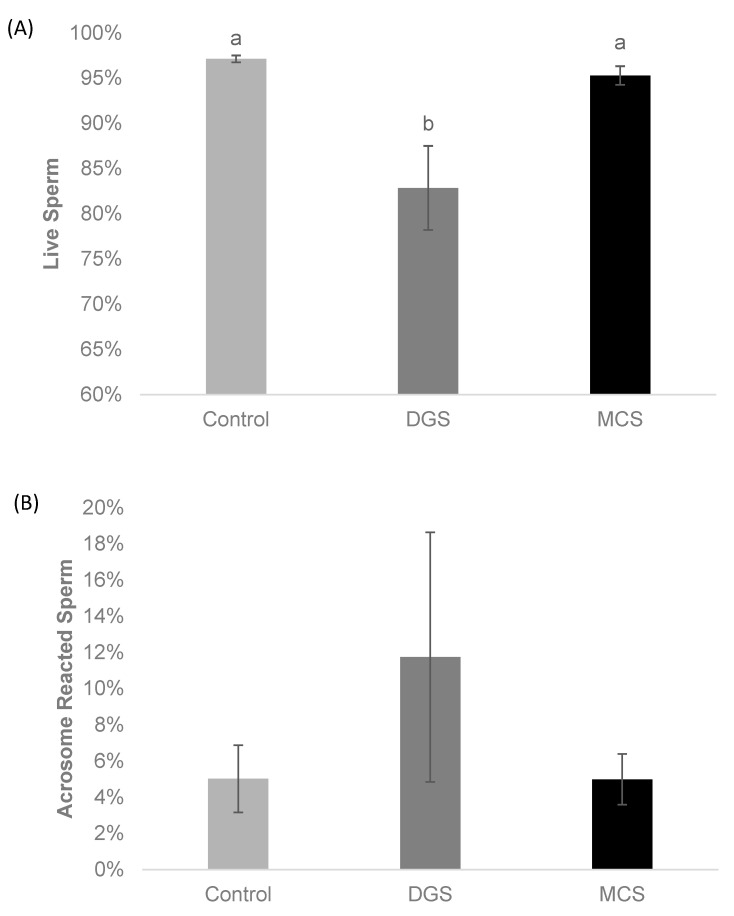
Sperm viability before (control) and after each sperm selection method (MCS, DGS). Data are shown as mean% ± SEM (N = 20 replicates for each group). (**A**) Proportion of sperm alive; Kruskal–Wallis Test (gl = 2; *p* < 0.01). (**B**) Sperm with acrosome reacted, Kruskal–Wallis Test (gl = 2; *p* = 0.18). a, b Different superscript letters indicate significant differences amongst the groups; groups that share the same letter are statistically indistinguishable from each other. Abbreviations: MCS = microfluidic chip-based sperm; DGS = density gradient system.

**Figure 5 animals-14-02544-f005:**
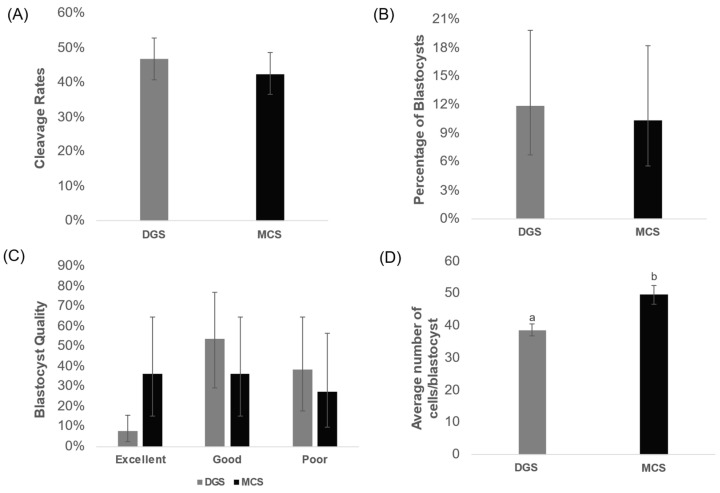
Effect of sperm selection methods on in vitro fertilisation outcomes and blastocyst quality. (**A**) Proportion of cleaved embryos after the use of sperm selection methods (DGS, MCS). Mean of the effects of each sperm selection method on cleavage rates after 3 days of embryo culture. Error bars show the 95% confidence interval. Groups did not differ (N = 430, *p* > 0.05). (**B**) Proportion of blastocyst formation after the use of both DGS and MCS. Mean of the effects of each sperm selection method on blastulation rates after 5 days of embryo culture. Error bars show the 95% confidence interval. Groups did not differ (N = 430, *p* > 0.05). (**C**) Effect of both sperm selection methods on blastocyst quality. Error bars show the 95% confidence interval (N = 48). Groups did not differ (*p* > 0.05). (**D**) Effect of sperm selection method on average number of cells per blastocyst. ANOVA Test (gl = 1, *p* > 0.05, F = 12.389). Error bars show the SEM (N = 48). a, b Different superscript letters indicate significant differences between the groups. Abbreviations: MCS = microfluidic chip-based sperm; DGS = density gradient system.

**Table 1 animals-14-02544-t001:** Proportion of each type of morphology anomalies in sperm before and after sperm selection.

Group	Head Abn. Mean% ± SEM	Tail Abn. Mean% ± SEM	CD Mean% ± SEM
Control	7.6 ± 0.7	10.5 ± 1.2 ^a^	11.5 ± 1.6 ^a^
MCS	6.7 ± 0.5	4.9 ± 0.6 ^b^	3.6 ± 0.6 ^b^
DGS	7.4 ± 0.5	8.6 ± 1.0 ^c^	8.8 ± 1.2 ^c^
*p* value *	0.402	<0.001	<0.001
H *	1.82	16.744	19.383

Data are shown as mean% ± SEM (N = 17 replicates for each group). Abbreviations: MCS = microfluidic chip-based sperm; DGS = density gradient system; Abn = abnormalities; CD = cytoplasmic droplet. * Kruskal–Wallis Test (gl = 2). ^a, b, c^ Within each column, different superscript letters indicate significant differences amongst the groups; samples that share the same letter are statistically indistinguishable from each other. H was calculated as variation between sample means/variation within the samples.

**Table 2 animals-14-02544-t002:** Effect of sperm preparation technique on % DNA fragmentation Index (DFI) for two sperm selection techniques: microfluidic chip-based sperm sorting (MCS) and density gradient system (DGS).

Group	DFI Mean% ± SEM	HDS Mean% ± SEM
Control	2.4 ± 0.3 ^a^	2.3 ± 0.4 ^a^
MCS	0.7 ± 0.1 ^b^	0.7 ± 0.1 ^b^
DGS	1.1 ± 0.2 ^b^	1.7 ± 0.3 ^a^
*p* value *	<0.001	<0.001
H *	27.76	27.53

Data are shown as mean% ± SEM (N = 20 replicates for each group). Abbreviations: MCS = microfluidic chip-based sperm; DGS = density gradient system; DFI = % DNA fragmentation Index; HDS = high DNA stainability. * Kruskal–Wallis Test (gl = 2). ^a, b^ Within each column, different superscript letters indicate significant differences amongst the groups; samples that share the same letter are statistically indistinguishable from each other. H was calculated as variation between sample means/variation within the samples.

## Data Availability

Data are contained within the article.

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
