# Peer review of "Effect of Two Different Sperm Selection Methods on Boar Sperm Parameters and In Vitro Fertilisation Outcomes"

_animals, 2024, doi:10.3390/ani14172544_

Round 1

Reviewer 1 Report

Comments and Suggestions for Authors

The manuscript “Effect of Two Different Sperm Selection Method on Basic Boar 2 Sperm Parameters and In Vitro Fertilization Outcomes” describes protocols for sperm selection for IVF in suine species.

The authors showed in the manuscript that two methods can be used with slightly different outcomes for boar sperm selection.

The paper has exciting results, and the methodology seems reliable. However, it needs some clarification before publication.

Major concerns

The results could be clearer in some areas of the manuscript. In the methodology, the authors don’t describe the initial sperm concentration in each protocol and are not consistent with the volume for sperm “resuspension.” However, in the results section, the sperm concentration is presented at sperm/mL, which makes the results inconsistent.

Ex.: In the DGS protocol, 1 mL of semen was processed and later the sperm was resuspended in 2 mL. Just this procedure has already reduced sperm/mL.

Please describe the initial sperm dose processed in each method and the total number of sperm recovered. For clarity, the authors could also present the sperm loss in percent or fold.

Statistical analysis must be reviewed. Several categorical variables were wrongly assessed as quantitative data (ANOVA and Kruskal-Wallis test).

The discussion needs to be greatly reviewed and improved before publication.

The conclusion is not supported and needs to be reviewed.

Figures must be improved. Figures can be merged in panels for clarity.

Specific points

L95 – When (months) was the study conducted? How were the semen samples maintained at room temperature? What exactly was the room temperature? How were the samples preserved at 17 °C?

L102 – How many sperm (sperm concentration) were used for processing?

L104 – The supernatant that was removed was the semen on top of the 35% BovinePure, or everything on top of the pellet on the bottom of the tube? Can this process be clarified, please?

L113 – What was the sperm concentration or sperm/mL used?

L130 – What PFA means? Did the authors check midpiece abnormalities?

L143—Could the authors extend the sperm concentration analysis? Was a Neubauer chamber used? If not, how was sperm concentration assessed using a Makler chamber? Software?

L146 – 166—Flow cytometry analysis needs to be expanded. Controls, filters, etc.… are missing. Please review this section.

L192 – Can the authors compare the number of days it took for embryos to reach the blastocyst stage between groups?

L218 – Statistical analysis must be reviewed.

Can the authors cite the normal and the non-parametric data in the text?

The number of cleavages, blastocysts, and embryo quality are a categorical variable (YES/NO per oocyte). Therefore, it can’t be assessed by ANOVA or Kruskal-Wallis. It must be reviewed.

L323-328 – it should be described in the methodology.

L378-389 – This paragraph is really confusing. Please rewrite and avoid “results”. Please discuss your findings instead.

Minor concerns

L 55 – “IVP”, please review. Is it “in vitro production,” or should it be IVF?

L 167 – What does IVM mean? In vitro maturation?! Please extend.

L197 - … oocytes were incubated with … “sperm”?!

L198-199 – “To reduce the risk of polyspermy, oocytes were moved to a clean well with PGM for another two hours.” – unclear sentence

L245 – Figures could be merged as a panel.

Figure 5 can be used as a supplement.

L303 – Please review the word “alive” throughout the manuscript. It should be “viable”. Some sperm can be motile and have a damaged plasma membrane.

L354-360—please review. The paragraph sounds like there was a difference, but in the end, there wasn’t. It isn't very clear.

Author Response

(Comments also present in the reviewer comments attachment).

Comment 1: The results could be clearer in some areas of the manuscript. In the methodology, the authors don’t describe the initial sperm concentration in each protocol and are not consistent with the volume for sperm “resuspension.” However, in the results section, the sperm concentration is presented at sperm/mL, which makes the results inconsistent.

Response: It has been presented as ML, because for the calculation of the concentration that information (ML of pgm used in each method) were also taken into account 

Comment 2: Ex.: In the DGS protocol, 1 mL of semen was processed and later the sperm was resuspended in 2 mL. Just this procedure has already reduced sperm/mL. 

Please describe the initial sperm dose processed in each method and the total number of sperm recovered. For clarity, the authors could also present the sperm loss in percent or fold.

Response: We showed the information of the samples before and after processing, so it is clear the effect that the selection method  

Comment 3: Statistical analysis must be reviewed. Several categorical variables were wrongly assessed as quantitative data (ANOVA and Kruskal-Wallis test. 

Response: Numerous publications have assessed these type of variables with anova and it is widely accepted. Here we attached some publications: 

Lloyd, R. E., Romar, R., Matás, C., Gutiérrez-Adán, A., Holt, W. V., & Coy, P. (2009). Effects of oviductal fluid on the development, quality, and gene expression of porcine blastocysts produced in vitro. Reproduction, 137(4), 679. 

Ferré, P., Bui, T. M. T., Wakai, T., & Funahashi, H. (2016). Effect of removing cumulus cells from porcine cumulus-oocyte complexes derived from small and medium follicles during IVM on the apoptotic status and meiotic progression of the oocytes. Theriogenology, 86(7), 1705-1710. 

Albal, M. S., Silvestri, G., Kiazim, L. G., Vining, L. M., Zak, L. J., Walling, G. A., ... & Griffin, D. K. (2022). Supplementation of porcine in vitro maturation medium with FGF2, LIF, and IGF1 enhances cytoplasmic maturation in prepubertal gilts oocytes and improves embryo quality. Zygote, 30(6), 801-808. 

Ritter, L. J., Sugimura, S., & Gilchrist, R. B. (2015). Oocyte induction of EGF responsiveness in somatic cells is associated with the acquisition of porcine oocyte developmental competence. Endocrinology, 156(6), 2299-2312. 

Comment 4: The discussion needs to be greatly reviewed and improved before publication. 

The conclusion is not supported and needs to be reviewed.

Response: The conclusion it is supported since the number of cells in MCS is significantly higher.

Comment 5: Figures must be improved. Figures can be merged in panels for clarity. 

Response: We do not consider that this comment will improve the understanding of the manuscript. 

Comment 6: L95 – When (months) was the study conducted? How were the semen samples maintained at room temperature? What exactly was the room temperature? How were the samples preserved at 17 °C?

Response: It was preserved in a cool box; this info has also been added to the manuscript

Comment 7: L102 – How many sperm (sperm concentration) were used for processing?

Response:  The concentration is calculated after the processing that’s why is describe in Ml.

Comment 8: L104 – The supernatant that was removed was the semen on top of the 35% BovinePure, or everything on top of the pellet on the bottom of the tube? Can this process be clarified, please? 

Response: All the supernatant was removed, so it has been added to the text.

Comment 9: L113 – What was the sperm concentration or sperm/mL used? 

Response:  Every sample had a different concentration, so before IVF the concentration was adjusted to the desired concentration (10^5 sperm/ml). 

Comment 10: L130 – What PFA means? Did the authors check midpiece abnormalities? 

Response: PFA is Paraformaldehyde and it has been added to the text. If any midpiece abnormality was found was included in tail abnormalities, because they were rare.

Comment 11: L143—Could the authors extend the sperm concentration analysis? Was a Neubauer chamber used? If not, how was sperm concentration assessed using a Makler chamber? Software?.

Response: As written in the Sperm concentration section the count was done using a Makler chamber. The formula used has been included in the text. Also, It has been added the concentration used for IVF.

Comment 12: L146 – 166—Flow cytometry analysis needs to be expanded. Controls, filters, etc.… are missing. Please review this section.

Response: Information added.

Comment 13: L192 – Can the authors compare the number of days it took for embryos to reach the blastocyst stage between groups?

Response: The embryos were not cultured in a Time Lapse system so they all were evaluated after 5 days. 

Comment 14: L218 – Statistical analysis must be reviewed.Can the authors cite the normal and the non-parametric data in the text?The number of cleavages, blastocysts, and embryo quality are a categorical variable (YES/NO per oocyte). Therefore, it can’t be assessed by ANOVA or Kruskal-Wallis. It must be reviewed. 

Response: It is already specified in each table and graph when the data was or wasn’t non parametric.  Numerous publications from have assessed  this type of variables with anova and it is widely accepted. Here we attached some publications: 

Lloyd, R. E., Romar, R., Matás, C., Gutiérrez-Adán, A., Holt, W. V., & Coy, P. (2009). Effects of oviductal fluid on the development, quality, and gene expression of porcine blastocysts produced in vitro. Reproduction, 137(4), 679. 

Ferré, P., Bui, T. M. T., Wakai, T., & Funahashi, H. (2016). Effect of removing cumulus cells from porcine cumulus-oocyte complexes derived from small and medium follicles during IVM on the apoptotic status and meiotic progression of the oocytes. Theriogenology, 86(7), 1705-1710. 

Albal, M. S., Silvestri, G., Kiazim, L. G., Vining, L. M., Zak, L. J., Walling, G. A., ... & Griffin, D. K. (2022). Supplementation of porcine in vitro maturation medium with FGF2, LIF, and IGF1 enhances cytoplasmic maturation in prepubertal gilts oocytes and improves embryo quality. Zygote, 30(6), 801-808. 

Ritter, L. J., Sugimura, S., & Gilchrist, R. B. (2015). Oocyte induction of EGF responsiveness in somatic cells is associated with the acquisition of porcine oocyte developmental competence. Endocrinology, 156(6), 2299-2312. 

Comment 15: L323-328 – it should be described in the methodology.

Response: Updated methods section ‘Sperm Viability and Acrosomal Integrity’ 

Comment 16: L378-389 – This paragraph is really confusing. Please rewrite and avoid “results”. Please discuss your findings instead.

Response: The paragraph has been modified

Comment 17: L 55 – “IVP”, please review. Is it “in vitro production,” or should it be IVF?

Response: Changed to IVF

Comment 18: L 167 – What does IVM mean? In vitro maturation?! Please extend. 

Response: Yes, the abbreviation has been clarified. 

Comment 19: L197 - … oocytes were incubated with … “sperm”?! 

Response: Sperm is now included in the sentence. 

Comment 20: L198-199 – “To reduce the risk of polyspermy, oocytes were moved to a clean well with PGM for another two hours.” – unclear sentence

Response: Now the sentences is “to reduce the risk of polyspermy, oocytes were then moved to a clean well of PGM, to reduce the concentration of sperm, for another two hours  

Comment 21: L245 – Figures could be merged as a panel. Figure 5 can be used as a supplement. 

Response: We do not consider that this comment will  improve the understanding of the manuscript.

Comment 22: L303 – Please review the word “alive” throughout the manuscript. It should be “viable”. Some sperm can be motile and have a damaged plasma membrane.

Response:  It has been changed to viable

Comment 23: L354-360—please review. The paragraph sounds like there was a difference, but in the end, there wasn’t. It isn't very clear.

Response:  We think the lines 360-366 are clearly describing what the graphic shows, which is that the percentage of excellent quality embryos is higher in MCS, but statistical analysis was not significant  

Reviewer 2 Report

Comments and Suggestions for Authors

The manuscript entitled: Effect of Two Different Sperm Selection Method on Basic Boar Sperm Parameters and In Vitro Fertilization Outcomes compares two methods of sperm sorting prior to in vitro production of embryos with the aim of generating better quality embryos.

The main objective of the study is to find a more practical solution compared to the established density gradient centrifugation. A microfluidic system is used for this purpose. This is easier to use and the use of chemicals (for example Percoll) can be completely dispensed with. 

The use of the iSperm system stands out in the methods of the study. This system is not recommended for collecting kinematic parameters due to the high variability of the measurement and the comparatively poor repeatability. I would strongly recommend using an established CASA system in the future. The system often reaches its limits, especially when measuring sperm samples with low concentrations. Even established CASA systems are often limited here. This could lead to a less meaningful interpretation of the kinematic parameters.

From line 368 there are some formatting errors that should be corrected before publication.

Overall, this is an interesting and scientifically correct study with a clear result. This is supported by the data shown.

Overall, I strongly recommend that the authors use an established CASA system in the future. This will make the kinematic data collected much more reliable. The manuscript can be published after minor revision.

Author Response

Comment: From line 368 there are some formatting errors that should be corrected before publication.

Response:  The formatting errors have been corrected

Round 2

Reviewer 1 Report

Comments and Suggestions for Authors

It is crucial that the authors address the major concerns with the manuscript, as this will significantly enhance the quality and credibility of the research.

The manuscript would greatly benefit from a clear and detailed methodology, as well as a thorough review of the statistical analysis. These are key areas that require attention and will significantly improve the quality of the research.

The authors don’t describe the initial sperm concentration in each protocol and are not consistent with the volume for sperm “resuspension.”, which makes the results inconsistent. 

Please describe the initial sperm dose processed in each method and the total number of sperm recovered. For clarity, the authors could also present the sperm loss in percent or fold.

Statistical analysis must be reviewed. Several categorical variables were wrongly assessed as quantitative data (ANOVA and Kruskal-Wallis test. 

The figures are poorly presented.

Author Response

Comment 1: It is crucial that the authors address the major concerns with the manuscript, as this will significantly enhance the quality and credibility of the research.

Response 1: We have thoroughly reviewed every comment and made the necessary changes on the document. We appreciate that the comments made have been very useful to better the manuscript. 

Comment 2: The manuscript would greatly benefit from a clear and detailed methodology, as well as a thorough review of the statistical analysis. These are key areas that require attention and will significantly improve the quality of the research.

Response 2: The statistical analysis has been reviewed and updated. Our methodology we have reviewed and updated accordingly.

Comment 3: The authors don’t describe the initial sperm concentration in each protocol and are not consistent with the volume for sperm “resuspension.”, which makes the results inconsistent. Please describe the initial sperm dose processed in each method and the total number of sperm recovered. For clarity, the authors could also present the sperm loss in percent or fold.

Response 3: The sperm concentration is talked about in results (e.g. Figure 2A) and method section. The key point is not that the dilution needs to be the same across methods. What we are evaluating here is the outcomes of the different sperm separation methods and how these relate to sperm parameters and IVF outcomes. The MCS and DGC methodology was done according to the manufacturers' guidelines. The focus is on comparing the end results and effectiveness of the different sperm separation techniques, rather than the specifics of the dilution process used in each case, although information on volume of sperm used and volume of media added is provided in the methods section. The important thing is assessing the overall performance and success rates of the various sperm separation approaches in relation to IVF success. Additionally, the presence of control allows the readers to know the concentration of sperm prior to sperm preparation. 

Comment 4: Statistical analysis must be reviewed. Several categorical variables were wrongly assessed as quantitative data (ANOVA and Kruskal-Wallis test. 

Response 4: This has been addressed in our manuscript and the statistical analysis section has been updated. 

Comment 5: The figures are poorly presented.

Response 5: We have updated the figures to make it into a panel. We believe this has been really beneficial for the manuscript to show the figures in that way.